# MAD: A Multimodal Anomaly Detection Framework Based on Shared Transformer and Contrastive Learning for Smart Manufacturing

## Abstract

With the advancement of smart manufacturing environments, anomaly detection techniques that integrate heterogeneous composite sensor data are becoming increasingly important. However, there are still technical difficulties in effectively fusing data with different characteristics, such as PRPD images and PD time series. To address these issues, this study proposes a high-performance multimodal framework, MAD, based on a two-step training strategy. First, to reduce the representation differences between modalities, a RealNVP-based normalization flow is introduced to align the representations of each modality into a shared latent space. Second, we use Supervised Contrastive Learning to learn a structured representation space with well-defined boundaries between classes. The aligned and structured representations are then fed into the LIMoE encoder, a Mixture-of-Experts-based shared transformer, to finally classify the types of anomalies. Experimental results demonstrate that the proposed MAD model outperforms the existing SOTA multimodal models. In particular, MAD achieves an AUC of 100.0% and F1-score of 99.98%, which is comparable to that of Perceiver IO, Cross-Modal Transformer, and CFM.

## 1 Introduction

In recent years, smart factories are being advanced through the convergence of artificial intelligence (AI) and information and communication technology (ICT), revolutionizing the entire manufacturing industry Wang et al. (2018). In particular, AI-based diagnostic systems are rapidly increasing in demand in the fields of predictive maintenance, quality control, and anomaly detection, which are emerging as key strategies for securing productivity and process reliability. Traditionally, single-sensor-based analytics has been the dominant approach, but recently, multimodal learning, which combines multiple data, has gained attention as a new alternative Mao & Zhang (2024). Multimodal learning enables more precise and reliable diagnosis by integrating the informational characteristics and strengths of different data, and when integrating various data such as images, time series, sound, and text, it can be expected to outperform single-modal-based models Qiao et al. (2025); Wu et al. (2024). However, learning between heterogeneous modalities with different physical characteristics such as time series signals and images is still challenging, and related research is relatively scarce Yuan et al. (2024); Nie et al. (2025); Xi et al. (2025). To address these issues, this study proposes a representation alignment-based unified latent space learning structure that can overcome the limitations of late fusion methods. The distribution differences between Phase Resolved Partial Discharge (PRPD) images and Partial Discharge (PD) time series signals are aligned using a Normalizing Flow-based alignment module to resolve distribution differences and mitigate semantic inconsistencies between modalities. This paper proposes a new framework, MAD (Multimodal Anomaly Detector), to effectively integrate heterogeneous multimodal data consisting of PRPD images and PD time-series signals for anomaly detection. The proposed framework is based on three main strategies. First, it aligns the differences in expression distributions between modalities using Real-valued Non-Volume Preserving transformation(RealNVP)-based normalizing flow. Second, it learns a representation space with clear class distinctions through supervised contrastive learning (SupCon)-based guided contrastive learning. Third, it applies a shared Transformer encoder based

on a mixture of experts (LIMoE) to effectively integrate heterogeneous representations and learn specialized features. Through this, this study makes the following contributions.

- We propose an integrated framework, MAD, that introduces a Normalizing Flow-based representation alignment module to resolve representation mismatch between heterogeneous modalities.
- We demonstrate the effectiveness of a two-stage learning strategy that combines SupCon-based pre-training with Cross-Entropy (CE)-based fine-tuning.
- We present an architecture that learns specialized features and integrates heterogeneous modalities without requiring large-scale parameters by leveraging a LIMoE-based shared encoder.
- We evaluate the performance of the proposed model through experiments using real-world industrial data and present comparative analysis against single-modality and state-of-the-art multimodal models.

The structure of this paper is as follows. Section 2 reviews related work. Section 3 describes the architecture and components of the proposed MAD framework. Section 4 presents experimental analyses to evaluate the performance of the proposed method. Finally, Section 5 concludes the paper and discusses future research directions.

## 2 RELATED WORKS

### 2.1 MULTI-MODAL TRAINING

Multimodal learning is an approach that integrates different types of data, such as images and time-series signals, to learn richer representations and improve model performance Bayoudh et al. (2022); Debie et al. (2021); Luo (2023). Data fusion methods are categorized into early fusion, which combines input data; late fusion, which aggregates prediction results; and mid-fusion, which integrates intermediate representations after extracting features from each data source Ma et al. (2021); Arifin et al. (2025). Among these, mid-fusion methods, which can directly learn interactions between modalities, have recently gained the widest adoption Qiu et al. (2024); Yu et al. (2024a).

### 2.2 NORMALIZING FLOW

Normalizing flows are a powerful class of generative models used for density estimation and latent space alignment. They transform complex data distributions into normalized latent spaces through a sequence of invertible functions Papamakarios et al. (2019). This approach enables the mapping of input data to interpretable latent representations, even for high-dimensional data with strong statistical nonlinearity, making it widely applicable in areas such as time-series forecasting and anomaly detection Kobyzev et al. (2021). Representative architectures include NICE Dinh et al. (2014), RealNVP Dinh et al. (2016), and Glow Kingma & Dhariwal (2018). These models construct multiple coupling layers to progressively transform the input information, and their invertible structure allows for both density estimation and efficient sampling.

### 2.3 LIMoE TRANSFORMER

LIMoE is a model that integrates the MoE mechanism into the Transformer architecture, efficiently allocating computational resources by selectively activating the optimal expert for each input token Yu et al. (2024b); Mustafa et al. (2022). The core of this model is the combination of a shared self-attention layer and a router that directs each token to the most suitable expert. This token-level expert selection approach enables effective processing of structurally diverse heterogeneous data within a single encoder Han et al. (2024).

### 2.4 CONTRASTIVE LEARNING

Contrastive learning is a representative self-supervised learning technique that learns discriminative embeddings by pulling similar samples closer and pushing dissimilar ones farther apart in the representation space Chen et al. (2020). Early contrastive learning methods, such as SimCLR He et al.

---

**Algorithm 1** Training procedure of MAD Model

---

**Input**: Dataset $(X, Y, L)$, where $X$ are PRPD image tokens, $Y$ are signal tokens, and $L$ are ground truth labels. Configuration parameters.
**Output**: Trained MAD model $M$.

1:  *// Stage 1: SupCon Pretraining*
2:  Initialize model $M$, optimizer, and scheduler
3:  **for** $epoch = 1$ to $N_{\text{pre}}$ **do**
4:      $f \leftarrow M(X, Y, \text{for\_supcon=True})$
5:      $\mathcal{L} \leftarrow \mathcal{L}_{\text{SupCon}}(f, L)$ {Supervised Contrastive Loss}
6:      Backpropagate loss and update model parameters
7:      Evaluate and save the best checkpoint
8:  **end for**
9:
10: *// Stage 2: CE Fine-tuning*
11: Load best weights from pretraining into $M$
12: **for** $epoch = 1$ to $N_{\text{fine}}$ **do**
13:     $z \leftarrow M(X, Y, \text{for\_supcon=False})$
14:     $\mathcal{L} \leftarrow \mathcal{L}_{\text{CE}}(z, L)$ {Cross Entropy Loss}
15:     Backpropagate loss and update model parameters
16:     Save the best fine-tuned model
17: **end for**
18: **return** $M$

---

(2019) and MoCo citeref26, focused primarily on image-based pretraining. With the advent of Sup-Con, it became possible to design contrastive losses that leverage label information. SupCon defines samples from the same class as positive pairs and those from different classes as negative pairs, thereby improving both intra-class compactness and inter-class separability. In multimodal settings, positive pairs can be formed across different modalities, or within the same modality through data augmentation or modality masking Dufumier et al. (2024). This approach effectively aligns cross-modal representations and enhances the semantic structure within the representation space.

# 3  MAD: MULTI-MODAL ANOMALY DETECTION

In smart manufacturing environments, integrating heterogeneous data such as PRPD images and PD time series to detect anomalies early is crucial, but simple fusion is ineffective. To address this issue, this study proposes the MAD anomaly detection framework, which performs multi-modal representation alignment and unified encoding. MAD consists of five stages: data preprocessing, normalization flow, shared transformer, contrastive learning, and classification. The overall structure is summarized in Figure 1, and the detailed learning procedure is outlined in Algorithm 1.

## A. DATA PREPROCESSING

To integrate the PRPD image and PD time series signal into a shared transformer, preprocessing was performed to convert each into a fixed-length sequence token. The PRPD image is a 256×256 RGB image divided into 16×16 patches according to the ViT method to generate a total of 256 tokens, each of which is embedded in 768 dimensions and then static positional encoding is added. The PD time series was converted to 2,560 tokens by dividing 20 seconds of data into 60-point increments and applying positional encoding to reflect temporal information. The converted sequences of both modalities are then merged into a normalization flow and shared encoder to learn a multimodal representation.

## B. NORMALIZING FLOW

To address representation mismatch between modalities, RealNVP-based normalizing flows are applied to each token sequence. This module maps the high-dimensional representations of both images and time-series data to a shared latent space by aligning them to a standard normal distribution,

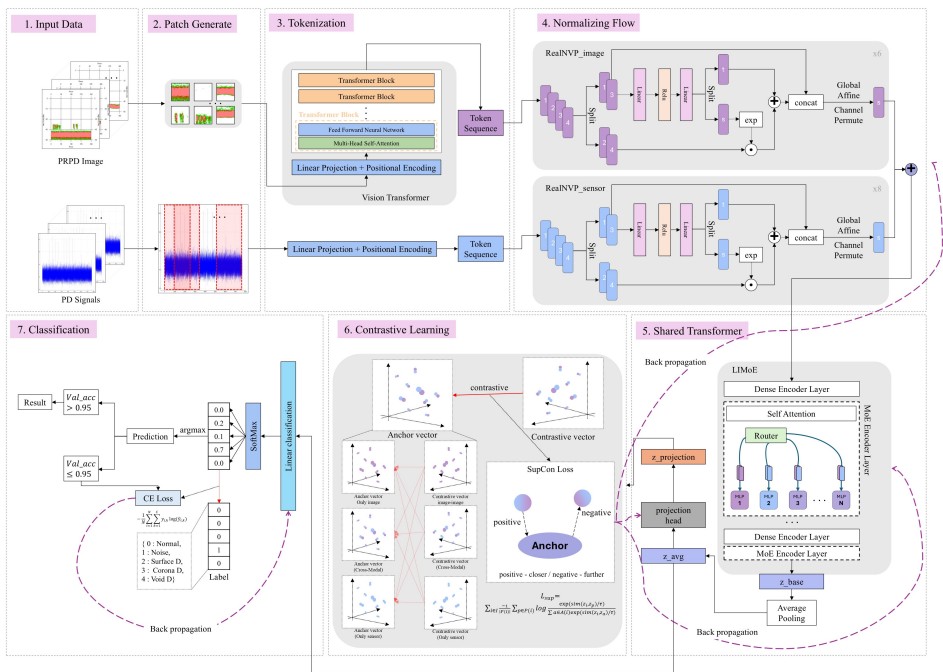

Figure 1: Overall architecture of the MAD framework. The model detects anomalies based on PRPD images and PD time-series signals through a five-stage pipeline: (A) Data Preprocessing, (B) Normalizing Flow, (C) Shared Transformer, (D) Contrastive Learning, and (E) Classification.

facilitating stable joint learning within the shared encoder. Specifically, the proposed model employs a RealNVPImageFlow composed of three coupling layers for image tokens, and a RealNVPSignalFlow composed of five causal coupling layers for time-series tokens. Each coupling layer adopts an affine transformation structure that splits the input into two parts—one part scales and shifts the other—enhancing model expressiveness through channel-wise permutations. This architecture results in a triangular Jacobian matrix, allowing for efficient training by simplifying the computation of the log-determinant of the Jacobian.

## C. SHARED TRANSFORMER

The images and time-series token sequences passed through normalizing flows are combined into a single input and fed into a shared Transformer encoder for unified representation learning. In this study, the encoder is based on the LIMoE architecture, which consists of four stacked encoder layers. Each layer comprises two sub-modules. The first module, Multi-Head Self-Attention, uses 8 attention heads to learn complex interactions between image and time-series modalities, thereby integrating cross-modal contextual information. The second module, Feedforward Network (FFN), adopts a MoE structure. The proposed model utilizes two expert MLP layers, simultaneously routing each token to both experts and integrating their outputs through ensemble. This approach aims to maximize the model's expressive power. Additionally, Pre-Normalization-based Layer Normalization is applied before each sub-module to enhance training stability and accelerate convergence speed.

## D. CONTRASTIVE LEARNING

Before commencing full classifier training, a pre-training step is performed to structurally align the embedding space and enhance the model's generalization performance. During this step, the SupCon loss is applied to encourage embeddings of the same class to cluster closely together while keeping different classes far apart. The learning procedure is as follows. First, the embeddings output from the LIMoE encoder are mapped to the embedding space for contrastive learning via the projection head. Subsequently, multiple views generated by augmenting the same sample within each mini-

batch are used to form contrasting pairs. The SupCon loss considers samples with the same label relative to a specific anchor as positive pairs, and samples with different labels as negative pairs. The loss function is defined to maximize similarity between positive pairs and minimize similarity between negative pairs. The temperature parameter, which controls the steepness of the similarity distribution, is set to 0.5.

The SupCon loss function is defined as follows:

$$\mathcal{L}_{\text{SupCon}} = \sum_{i \in I} \frac{-1}{|P(i)|} \sum_{p \in P(i)} \log \frac{\exp(\mathbf{z}_i \cdot \mathbf{z}_p / \tau)}{\sum_{a \in A(i)} \exp(\mathbf{z}_i \cdot \mathbf{z}_a / \tau)} \tag{1}$$

where $\mathbf{z}_i$ and $\mathbf{z}_p$ are the normalized projection vectors of the anchor and a positive sample, $P(i)$ is the set of positives for anchor $i$, and $A(i)$ is the set of all other samples in the batch excluding $i$.

Through this pretraining process, the model goes beyond merely acting as a classifier—it internalizes both cross-modal representation alignment and inter-class discrimination structure. This establishes a strong foundation for the subsequent fine-tuning stage, enabling the model to achieve high classification performance with minimal additional training.

### E. CLASSIFICATION

Based on a structurally aligned representation space obtained through pre-training, fine-tuning is performed for the final classification task. In this step, the projection head used in pre-training is removed, and a new classification head is attached. The classification head consists of a linear classifier that applies average pooling to the output sequence of the LIMoE encoder to summarize it into a single vector, which is then used as input for prediction. This structure aims to achieve high classification performance while maintaining lightweightness. The model is trained to distinguish between five classes (Normal, Noise, Surface, Corona, Void) and is optimized based on CE Loss.

## 4 EXPERIMENT AND RESULTS

### 4.1 EXPERIMENTAL SETUP

This study utilized a dataset of PRPD images and PD time-series signals collected from the AI Hub, synchronized in a 1:1 ratio. The data comprises five classes, including 'Normal', with a total of 60,020 samples distributed evenly. For fair performance evaluation, the entire dataset was split 1:1 into training and test sets. The input data leverages both spatial distribution (PRPD) and temporal pattern (PD) features (Figure 2). Model training was performed in a PyTorch environment, with the random seed fixed at 42 for reproducibility. The AdamW optimizer with a warm-up and cosine annealing scheduler was used, and the learning rate was set to $5 \times 10^{-6}$. To ensure stable training, gradient clipping and an early stopping mechanism monitoring the validation F1-score were applied. The full experimental setup is summarized in Table 1.

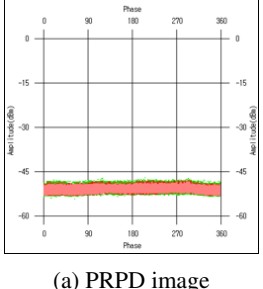

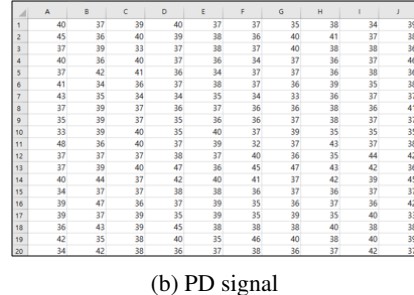

(a) PRPD image                                    (b) PD signal

Figure 2: Examples of multimodal input data: (a) PRPD image and (b) PD signal.

Table 1: Summary of experimental settings and follow-up experiments.

| Item | Description |
|---|---|
| Dataset | PRPD images + PD time-series |
| Classes | 5 types (N, S, C, V, etc.) |
| Dist. | Uniform 20% per class |
| Framework | PyTorch |
| Hardware | NVIDIA RTX 4090 GPU |
| LR | $5 \times 10^{-6}$ |
| Optimizer | AdamW |
| LR Sched. | Warm-up + Cosine Annealing |
| **Regularization** | **Gradient Clipping (threshold=1.0)** |
| | **Early Stopping (patience=20)** |
| Experiments | • Multimodal vs. Single Modality |
| | • Model Size Effect |
| | • Impact of Loss Function |
| | • Baseline Comparison |

## 4.2 RESULTS

### 4.2.1 STRUCTURING FEATURE SPACE VIA MODALITY ALIGNMENT

This section visually validates the effectiveness of the proposed RealNVP-based Normalizing flow module. Figure 3 illustrates how feature vectors extracted from heterogeneous modalities are transformed through semantic alignment, the core process of our model. As seen in Figure 3(a), prior to transformation, data points from each class are intricately entangled within the latent space, with unclear boundaries between classes. This indicates that using unaligned features directly makes it difficult for the classification model to clearly distinguish between classes. However, after undergoing the Normalizing Flow, as shown in Figure 3(b), the latent space is reconstructed into a much more structured form. Specifically, data points belonging to the same class cluster closely together, increasing intra-class compactness, while the distance to other classes increases, significantly improving inter-class separability. These results suggest that the proposed Normalizing Flow module effectively aligns the distributions of heterogeneous modalities, enabling the subsequent Transformer encoder to successfully learn a meaningful feature space that allows for easier and more accurate classification of object types.

### 4.2.2 MULTI-MODAL VS. SINGLE-MODAL

This experiment aims to quantitatively analyze the effectiveness of the multimodal fusion structure by comparing the representation learning and classification performance of the proposed MAD with unimodal models. For this purpose, we constructed three models with the same basic structure but different input modalities, and the model summaries are presented in Table 2. Each model was sequentially subjected to SupCon-based pretraining and CE Loss-based fine-tuning. After pretraining, the multimodal model had a SupCon loss of 7.38, which is significantly lower than the image-only (18.64) and time-series-only (18.63) models. This indicates that the integrated modality learning forms a more structured embedding space. In the final classification performance, both the multimodal model and the image-only model demonstrated nearly perfect performance, recording high F1-scores of 0.9997 and 0.9999, respectively. This result indicates that both models achieved a similar top-tier level of performance, while also suggesting that the current dataset is heavily reliant on image information. In contrast, the time-series model showed the lowest performance at 0.9516, highlighting the limitations of using time-series information alone. On the other hand, the final classification performance after fine-tuning showed that the image-only model slightly outperformed the multimodal model (0.9997) with an F1-score of 0.9999, while the time series model was the lowest at 0.9516. This suggests the bias of the dataset toward image information and the limitations

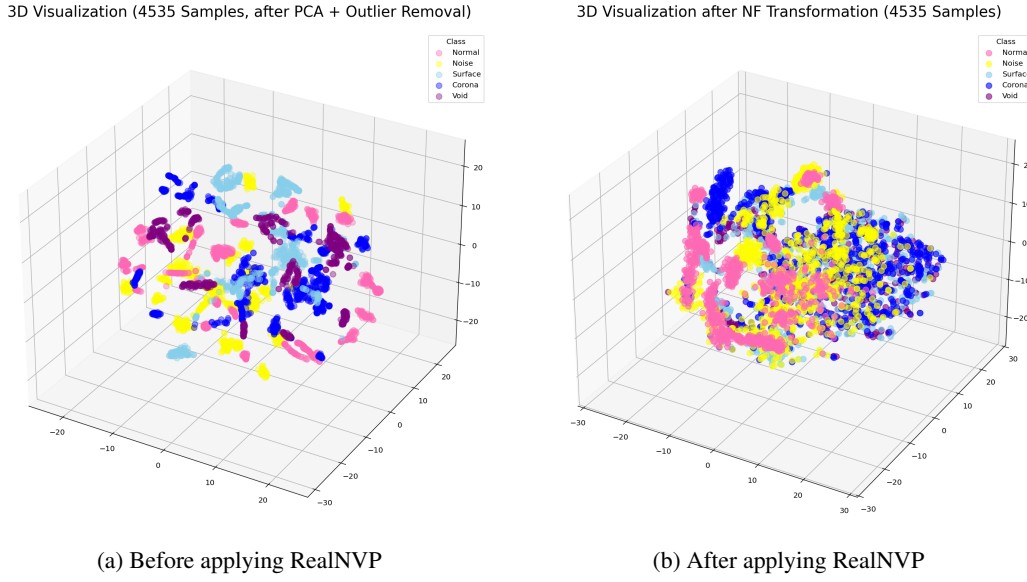

(a) Before applying RealNVP

(b) After applying RealNVP

Figure 3: Distribution comparison before and after modality alignment using RealNVP.

Table 2: Model variants and their configurations. I/S denotes the number of coupling layers for Image/Signal in RealNVP. D/H/E indicates the Depth, number of Heads, and number of Experts in the LIMoE encoder. Pre indicates whether SupCon pre-training was applied (O: Yes).

| Model | Modality | RealNVP (I/S) | LIMoE (D/H/E) | Pre |
|---|---|---|---|---|
| MAD (BASE) | Img+Sig | 3 / 5 | **4 / 8 / 1** | O |
| MAD_img_only | Img only | 3 / 0 | **4 / 8 / 1** | O |
| MAD_sig_only | Sig only | 0 / 5 | **4 / 8 / 1** | O |

of time series information. In the confusion matrix comparison (Figure 4), the multimodal model (a) performed well with almost complete diagonalization, and the image-only model (b) performed similarly. On the other hand, the time series model (c) showed high misclassification rates for some classes. This shows that time series information alone is limited for classification.

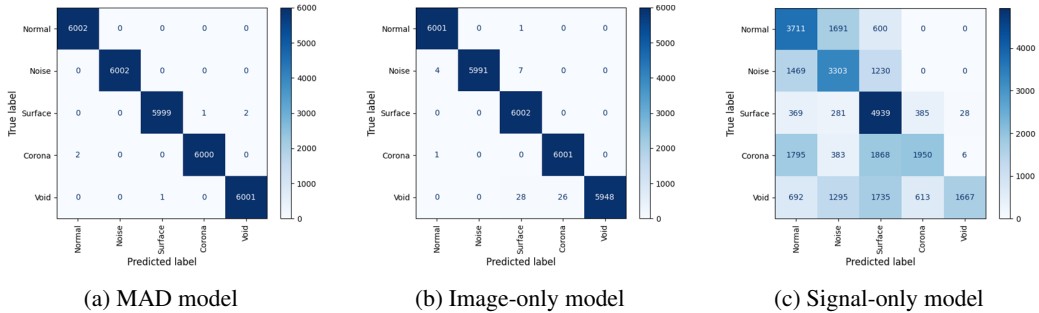

(a) MAD model

(b) Image-only model

(c) Signal-only model

Figure 4: Comparison of confusion matrices on the test set for models trained with (a) multi-modal input, (b) image-only input, and (c) signal-only input.

### 4.2.3 MODEL SIZE EFFECT

In this section, we compare a large model (MAD_base) and a lightweight model (MAD_small) to analyze the impact of model size and complexity on training stability and performance. The two models had 75.77M and 2.38M parameters, respectively, and followed the same training procedure.

Table 3: Comparison of Model Size and Performance between MAD_base and MAD_small.

| Model | #Params (M) | GFLOPs | F1-score (%) | Validation Stability |
|---|---|---|---|---|
| MAD_base | 75.77 | 318.53 | 99.98 | Unstable |
| MAD_small | 2.38 | 2.10 | 99.50 | Stable |

In the learning curve (Figure 5), MAD_base showed large fluctuations in validation performance and unstable convergence, while MAD_small showed a smooth and stable learning pattern. Despite its training instability, MAD_base ultimately achieved a superior F1-score of 99.98%, as shown in the performance and complexity table 3. MAD_small, while achieving a slightly lower F1-score, offers a more robust learning process and significantly better computational efficiency (GFLOPs). Therefore, in real-world applications, there is a clear trade-off: choosing MAD_base for maximum performance, or MAD_small for stability and efficiency.

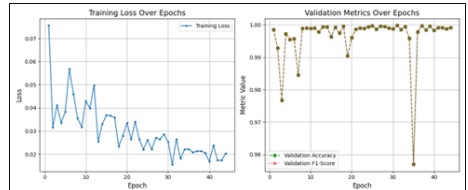 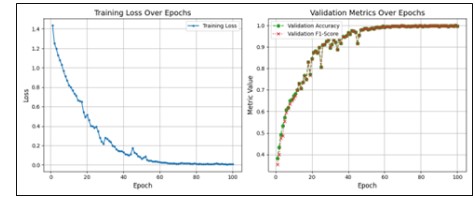

(a) MAD_base: training loss (left), validation accuracy and F1-score (right)

(b) MAD_small: training loss (left), validation accuracy and F1-score (right)

Figure 5: Comparison of loss curves on the test set for (a) MAD_base and (b) MAD_small.

### 4.2.4 IMPACT OF LOSS FUNCTION

In this section, we analyze the impact of the loss function design, specifically the application of SupCon-based dictionary learning, on classification performance. Two models are compared (a) a two-stage learning model with SupCon pretraining followed by CE-based fine-tuning, and (b) a model with end-to-end learning with CE loss alone. The performance was evaluated via confusion matrices on the test set, and the results are presented in Figure 6. The experimental results show that the model (a) with SupCon-based learning has overall balanced classification performance and low misclassification rate. On the other hand, the CE-only model (b) showed high recall in some classes (e.g., 'Normal'), but tended to over-predict other classes to that class, resulting in poor precision. This misclassification was particularly noticeable for the 'Corona' class. By learning the boundaries between classes more clearly, the model using SupCon provided higher precision and more stable predictions. This means that the model makes more cautious predictions and maintains high confidence in each prediction. Therefore, considering the reliability of predictions and the balance between classes in addition to simple accuracy, the SupCon-based two-stage learning strategy is a more effective and practical approach for anomaly detection systems.

### 4.2.5 BASELINE COMPARISON

To evaluate the performance of the proposed MAD model, comparative experiments were conducted with representative SOTA multimodal models. For the comparison, the test dataset, consisting of about 30,000 synchronized image-time series data of five classes, was used, and all models were trained and evaluated under the same conditions. The baselines include CFM, CMT, and Perceiver IO, which feature memory-based delay fusion, ViT-cross-attention coupling, and modality-independent latent space fusion structures, respectively. As summarized in Table 4, the proposed MAD model achieved an F1-score of 99.98%, a performance highly comparable to top-tier models like CMT (99.99%) and CFM (100.0%). This result demonstrates that our proposed strategy, which combines normalizing flows for alignment and a SupCon-trained expert encoder, is a highly effective approach for multimodal anomaly detection.

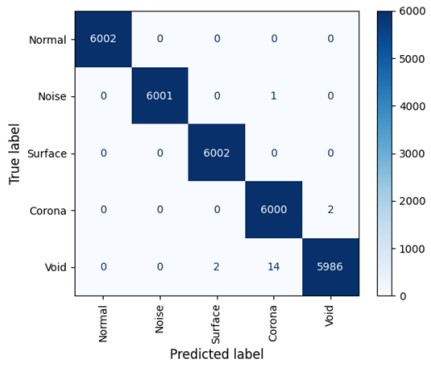 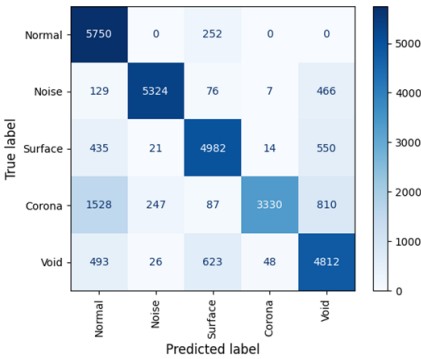

(a) SupCon + CE (2-stage training)       (b) CE only (end-to-end training)

Figure 6: Comparison of confusion matrices on the test set for models trained using (a) SupCon + CE and (b) CE only.

Table 4: Performance comparison with SOTA models. *CMT is an abbreviation for Cross-Modal Transformer.

| Model | AUC / F1 (%) | Key Features |
|---|---|---|
| Perceiver IO | 100.0 / 99.46 | Latent-space cross-attention |
| CMT* | 100.0 / 99.99 | ViT + Transformer + Cross-Attention |
| CFM | 100.0 / 100.0 | Cross-Fusion Memory |
| **MAD (Ours)** | **100.0 / 99.98** | RealNVP + LIMoE + SupCon |

## 5 CONCLUSION

In this work, we propose an integrated learning framework, MAD, to detect anomalies by integrating heterogeneous multimodal data consisting of PRPD images and PD time series signals. The proposed model uses a RealNVP-based normalization flow to align cross-modality representations, SupCon-based contrastive learning to learn a structured representation space, and a shared transformer encoder with MoE structure for final classification. Experimental results show that the proposed two-stage learning strategy (SupCon + CE) has higher classification performance and generalization ability than the single CE loss alone, and is more robust, especially for difficult classes. In the model lightweighting experiments, MAD_base achieved the highest accuracy, while MAD_small reduced the computation by more than 99% while maintaining stable learning and good performance. This suggests that the balance between accuracy and computational efficiency is important in real-time applications. In the future, we plan to introduce model compression techniques such as knowledge distillation, quantization, and pruning to further improve lightweight and real-time processing performance without sacrificing performance.

**Key contributions include:**

- Propose an integrated framework, MAD, featuring a Normalizing Flow-based module for aligning heterogeneous modality representations.

- Apply a two-stage learning strategy, combining SupCon pre-training and CE fine-tuning, to effectively structure the representation space.

- Present a LIMoE-based shared encoder architecture that integrates heterogeneous modalities and learns specialized features without excessively large parameters.

- Provide a comparative analysis of the proposed model against single-modality and recent multi-modal models using real-world industrial data.

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

## A  APPENDIX

Portions of this paper were written with the assistance of Google's large language model, Gemini. The model was used for tasks such as improving clarity and conciseness, correcting grammatical errors, and restructuring paragraphs based on reviewer feedback. The authors retained full responsibility for the final content and all scientific claims presented in this work.

