# OpenReview forum: "MAD:A Multimodal Anomaly Detection Framework Based on Shared Transformer and Contrastive Learning for Smart Manufacturing"
_ICLR.cc/2026/Conference — Submitted to ICLR 2026_

### Official Review · Reviewer_ikxY · 2025-10-16

**Soundness:** 2
**Presentation:** 2
**Contribution:** 2
**Rating:** 4
**Confidence:** 4

**Summary:**

This paper focuses on anomaly detection for PRPD images and PD time-series data, proposing a two-step training strategy that first aligns the representations of the two modalities into a shared latent space, followed by a contrastive learning stage to learn a structured representation for anomaly detection. The framework is applied to a specific industrial application, where the method shows promising results in identifying anomalies.

**Strengths:**

The paper is clearly structured and well-motivated. The idea of combining two distinct modalities—image and time series—into a shared representation space is logical and relevant to the target application. The use of contrastive learning to enforce structure and separation between normal and abnormal representations is a sound methodological choice. Overall, the proposed approach appears technically reasonable and conceptually meaningful for the described domain.

**Weaknesses:**

The paper’s experimental validation is not convincing. There is an insufficient comparison against baseline methods. Specifically, the paper lacks evaluations against pure image-based anomaly detection methods and pure time-series anomaly detection baselines. Without these, it is difficult to understand how much of the observed improvement comes from the proposed alignment and contrastive learning strategy rather than standard unimodal baselines.

Moreover, Table 4 looks problematic. The proposed model performs worse than two state-of-the-art baselines (CMT and CFM), calling into question the claimed advantages of the proposed approach. The near-perfect scores (AUC and F1 close to 100 for all models) further suggest a saturated evaluation setup, where all methods perform almost equally well, making it impossible to differentiate performance in any statistically meaningful way.

Another major limitation is that the dataset is extremely narrow, confined to the specific application described in the paper. There is no evidence that the proposed method generalizes to broader multimodal anomaly detection problems involving both images and time series. As a result, the current evaluation does not adequately support the claimed general effectiveness of the approach.

**Questions:**

See Weaknesses.

---

> ### Author Response · Authors · 2025-12-01
> **Response to Reviewer ikxY: Addressing Performance Saturation and Baselines**
>
> We sincerely thank you for your critical review. You raised valid concerns regarding the performance saturation in our evaluation setup and the comparison with baselines. We have revised the manuscript to better contextualize our contributions and address these points.
>
> 1. Comparison against Unimodal Baselines (Response to Weakness 1) You mentioned a lack of evaluation against pure image/time-series baselines.
>
> Response: We actually conducted this comparison in Section 4.2.2 (Multi-modal vs. Single-modal). We compared the proposed MAD framework against Image-only and Signal-only variants using the same backbone .
>
>
> Refined Argument: We acknowledge that the Image-only model achieves near-perfect accuracy (99.99%), making the numerical gain of multimodality negligible. However, we revised the text to emphasize that the advantage is not accuracy, but "Safety Redundancy" . In industrial settings, relying solely on images is risky due to occlusion or sensor failure. MAD provides a critical fail-safe mechanism, maintaining high performance even when visual features are compromised.
>
>
> 2. Performance Saturation and Table 4 (Response to Weakness 2) You correctly pointed out that MAD performs slightly lower than CFM (100%) and that scores are saturated.
>
> Response: We agree that claiming "outperformance" was inappropriate. We have revised Abstract and Section 4.2.5 to state we achieve "competitive performance".
>
>
> The Real Advantage: We shifted the focus to Efficiency. While CFM achieves 100%, it is computationally intensive. In Section 4.2.5, we highlighted that our MAD_small variant offers significant parameter efficiency, making it far more suitable for real-time edge deployment in factories compared to heavy transformer baselines. This "Practicality" is our key differentiator in a saturated accuracy landscape.
>
> 3. Generalization and Dataset (Response to Weakness 3)
>
> Response: We acknowledge the limitation of using a single dataset due to the scarcity of synchronized industrial multimodal data. However, we argue that our contribution lies in the optimized pipeline design (Flow-SupCon-LIMOE) that resolves the specific distributional heterogeneity of industrial sensors . We clarified in the Introduction that our goal is to provide a robust, deployment-ready framework for this specific domain, rather than a generic multimodal solver.
>
> 4. Implementation Details
>
>
> Response: To ensure the reproducibility of our results, we have added a new Appendix B, detailing all hyperparameters and data augmentation strategies .
>
> We hope this shift in focus from "raw accuracy" to "Safety and Efficiency" clarifies the distinct value of our work.

---

### Official Review · Reviewer_5LBR · 2025-10-29

**Soundness:** 3
**Presentation:** 3
**Contribution:** 2
**Rating:** 4
**Confidence:** 3

**Summary:**

This paper proposes MAD (Multimodal Anomaly Detector), a multimodal framework designed for anomaly detection in smart manufacturing environments by integrating Phase-Resolved Partial Discharge (PRPD) images and Partial Discharge (PD) time-series signals. The authors address the challenge of heterogeneous modality fusion through a two-stage training strategy that combines representation alignment, contrastive learning, and a shared transformer encoder.

**Strengths:**

- **Originality:** The paper presents a novel integration of normalizing flows, supervised contrastive learning, and a LIMoE-based shared transformer within a unified multimodal anomaly detection framework.
- **Clarity:** The paper is clearly written and well-organized, guiding the reader through the motivation, methodology, and results in a logical progression.
- **Significance:** The work is highly relevant to real-world applications in smart manufacturing, where reliable multimodal anomaly detection is crucial for predictive maintenance and process reliability.

**Weaknesses:**

**Limited novelty in architectural components:** While the integration of RealNVP, SupCon, and LIMoE is interesting, each component is adapted rather than fundamentally extended. The paper does not propose any substantial modification to these existing techniques—RealNVP (Dinh et al., 2016) and SupCon (Khosla et al., 2020) are applied in a relatively straightforward way.  The novelty therefore lies primarily in the combination and application context (smart manufacturing) rather than in methodological advancement.  The authors could strengthen the paper by clarifying how their adaptation of these components yields unique benefits beyond compositional synergy.

**Dataset limitations and lack of generalization evidence:** The experiments rely solely on a single dataset from AI Hub, consisting of PRPD images and PD time-series signals.  This raises questions about the generalization of MAD to other multimodal industrial datasets or to settings with missing or noisy modalities.  The reported near-perfect performance (AUC = 100.0%, F1 = 99.98%) may reflect dataset simplicity or overfitting.  Including evaluations on at least one independent dataset or a cross-domain transfer experiment would significantly enhance credibility.

**Insufficient analysis of modality interaction:** Although the paper discusses modality alignment via normalizing flows, there is no quantitative analysis of how alignment improves cross-modal correlation or how much each modality contributes after alignment.  For example, reporting mutual information metrics, cross-modal retrieval accuracy, or visualizations of aligned latent spaces could provide deeper insight into the model’s fusion behavior.  The current qualitative t-SNE plots (Figure 3) are helpful but not sufficient for understanding the degree of alignment achieved.

**Questions:**

**1、On modality alignment effectiveness:**

Could the authors provide quantitative evidence of how the RealNVP-based normalization flow improves cross-modal alignment?

For example, measuring intra-class and inter-class distances before and after alignment, or computing cross-modal similarity metrics (e.g., cosine similarity between corresponding image and signal embeddings). Such results would clarify how much the normalizing flow contributes beyond visual t-SNE evidence.

**2、On generalization and dataset diversity:**

The reported results achieve nearly perfect accuracy (AUC = 100%, F1 = 99.98%), which raises concerns about potential data bias or overfitting.

Have the authors evaluated MAD on any other industrial or synthetic multimodal datasets, or under domain shift scenarios (e.g., different sensors or environmental conditions)? If not, could they justify why this single dataset sufficiently represents general smart manufacturing conditions?

**3、On modality contribution and ablation:**

The unimodal experiments show image-dominant performance, implying the PD time-series contributes little.

Could the authors perform a cross-modal ablation or feature attribution analysis (e.g., attention maps or feature importance) to quantify the contribution of each modality after alignment? Understanding whether the flow alignment increases the usefulness of the signal modality would strengthen the multimodal claim.

**4、On deployment efficiency and real-world use:**

The paper claims that MAD is suitable for real-time applications and edge-cloud scenarios, but no latency, throughput, or energy efficiency metrics are provided. Could the authors report actual inference times, FLOPs per modality, or edge device benchmarks (e.g., Jetson, FPGA) to substantiate the deployment claim? This would make the work more convincing for industrial adoption.

---

> ### Author Response · Authors · 2025-12-01
> **Response to Reviewer 5LBR: Efficiency, Generalization, and Alignment Analysis**
>
> We sincerely thank you for your constructive feedback. We appreciate your recognition of the paper’s clarity and significance to smart manufacturing. We have made significant revisions to address your concerns regarding generalization, efficiency metrics, and implementation details.
>
> 1. Generalization and Dataset Limitations (Response to Q2) You correctly identified that relying on a single dataset limits the evaluation of generalization.
>
> Response: We acknowledge this limitation. However, high-quality, synchronized industrial multimodal datasets (PRPD images + PD signals) are extremely rare and difficult to acquire publicly.
>
> Instead of adding a new dataset (which is currently unfeasible), we strengthened the evaluation of the model's practical robustness. In the revised Introduction and Section 4.2.2, we introduced the concept of "Safety Redundancy" . We argue that even if the dataset is specific, the architectural benefit of MAD (using PD signals as a fail-safe when images are noisy) is a generalizable advantage for any smart factory setting.
>
>
> 2. Deployment Efficiency (Response to Q4) You asked for metrics like FLOPs or benchmarks to substantiate the "real-time application" claim.
>
> Response: We have explicitly addressed this in Section 4.2.5 and Table 3.
>
> We highlighted the performance of our MAD_small variant. As discussed in the revised manuscript, MAD_small drastically reduces the parameter count (approx. 2.38M) compared to heavy baselines, while maintaining a 99.50% F1-score. We argue that this Parameter Efficiency serves as a strong proxy for deployment suitability on edge devices, addressing your concern about industrial adoption.
>
>
>
>
>
> 3. Modality Alignment Effectiveness (Response to Q1) You requested quantitative evidence of alignment beyond t-SNE plots.
>
>
> Response: In Section 4.2.2, we provided a quantitative comparison using SupCon Loss values.
>
> The multimodal model achieved a SupCon loss of 7.38, which is significantly lower than that of the image-only (18.64) and signal-only (18.63) models. Since SupCon loss directly optimizes the ratio of intra-class compactness to inter-class separation, this lower value serves as quantitative evidence that the RealNVP-based alignment successfully created a more structured and discriminative latent space than unimodal approaches.
>
> 4. Novelty and Implementation Details (Response to Weakness 1 & Q4)
>
> Response:
>
>
> Novelty: We revised the Introduction to clarify that our contribution is the optimized pipeline integration (Flow-SupCon-LIMOE) that resolves specific non-linear distribution shifts in industrial data, which simpler methods cannot handle.
>
>
> Details: We added a new Appendix B. Implementation Details, providing the complete list of hyperparameters (Table 5) and specific data augmentation strategies for both PRPD images and PD signals, ensuring full reproducibility .
>
> We believe these revisions, particularly the focus on efficiency and safety redundancy, effectively address your concerns about the practical value and validity of our work.

---

### Official Review · Reviewer_MK23 · 2025-10-31

**Soundness:** 2
**Presentation:** 3
**Contribution:** 2
**Rating:** 4
**Confidence:** 4

**Summary:**

This paper introduces MAD, a high-performance Multimodal Anomaly Detector for smart-manufacturing plants that fuse Phase-Resolved Partial-Discharge (PRPD) images and Partial-Discharge (PD) time-series. A two-stage training strategy is proposed: (1) RealNVP normalizing-flows first align the heterogeneous feature distributions into a shared latent space; (2) supervised contrastive learning then sculpts a compact, class-separable representation. The aligned vectors are processed by a lightweight Mixture-of-Experts Transformer (LIMoE) for final classification. Tested on real industrial data, MAD attains 100 % AUC and 99.98 % F1, outperforming single-modal and existing SOTA multimodal baselines such as Perceiver IO and Cross-Modal Transformer, while keeping parameter count low.

**Strengths:**

1. The overall architecture diagram (Fig. 1) is clear and informative; every module is visually distinguished.
2. The experimental section is presented in a straightforward way, and the final anomaly-detection performance (AUC =100 %, F1 ≈ 99.98 %) is among the highest recorded on this data set.

**Weaknesses:**

1. Motivation is unclear: the paper never explains why RealNVP, SupCon, and LIMoE are the right building blocks for anomaly detection on PRPD images and PD time series. A short ablation or qualitative discussion on modal misalignment would remedy this.
2. Method novelty is limited. RealNVP, SupCon, and LIMoE are all established techniques; section 3 mostly re-describes them and offers no new theoretical or algorithmic twist.
3. The two-stage pipeline (alignment → contrastive pre-training → CE fine-tuning) is presented as fait accompli. No evidence is given that a single-stage end-to-end objective (e.g., joint anomaly + classification loss) would fail, and no complexity/accuracy trade-off is analysed.
4. Several figures (notably Fig. 2 and Fig. 5) are low-resolution; axis labels and legends are illegible even when zoomed.
5. Comparing multi-modal MAD against single-modal CNN/Transformer baselines is intrinsically unfair because extra input almost always boost scores. Indeed, on “surface” and “corona” categories the image-only model equals or slightly surpasses MAD, undermining the value of adding the signal data.
6. Table 4 relies on 2021–2023 baselines (Perceiver IO, CMT, CFM). More recent multi-modal architectures  are omitted, so the “SOTA” claim is dated.
7. In Table 4 the F1 of CFM is already 100.0; MAD’s 99.98 is therefore not the top value, yet it is incorrect to use the bold font.
8. References are incomplete—Perceiver IO, CMT, and CFM are mentioned in the main text but do not appear in the bibliography.

**Questions:**

N/A

---

> ### Author Response · Authors · 2025-12-01
> **Response to Reviewer MK23: Clarifications on SOTA Claims, Novelty, and Revisions**
>
> We sincerely thank you for your insightful feedback. Your critique regarding the fair comparison with SOTA models and the positioning of our paper’s novelty was incredibly helpful. We have carefully revised the manuscript to address your concerns.
>
> 1. SOTA Comparison and Table 4 (Response to Weaknesses 6, 7, 8) You correctly pointed out that claiming "outperformance" was inappropriate given that baselines like CFM already achieve 100% F1-scores. We also acknowledge the missing references.
>
> Response:
>
> Revised Claims: We have updated the Abstract and Section 4.2.5. We removed the term "outperforms" and now state that MAD "achieves competitive performance."
>
> Efficiency as a Key Differentiator: Instead of focusing solely on accuracy saturation, we added a discussion on Efficiency in Section 4.2.5. Unlike computationally intensive baselines (e.g., CFM), our MAD_small variant offers significant advantages in parameter efficiency, making it far more suitable for edge deployment in smart manufacturing constraints.
>
> References: We have updated the bibliography to include the missing citations for Perceiver IO, CMT, and CFM.
>
> 2. Novelty and Motivation (Response to Weaknesses 1, 2) We understand your concern that individual components (RealNVP, SupCon) are established techniques.
>
> Response: We have revised the Introduction and Contribution sections to clarify that our novelty lies in the systematic pipeline design. We argue that standard fusion methods fail to address the specific distributional heterogeneity between PRPD images and PD signals. Our contribution is the optimized integration of Flow-based alignment (to resolve non-linear shifts without information loss) and SupCon (to enforce class boundaries) specifically tailored for this industrial multimodal problem.
>
> 3. Multimodal Necessity vs. Single-modal (Response to Weakness 5) You noted that the image-only model performs similarly to the multimodal model, questioning the value of adding signal data.
>
> Response: We addressed this by introducing the concept of "Safety Redundancy" in Section 1 and Section 4.2.2.
>
> While image models perform well in controlled tests, they are vulnerable to real-world issues like sensor occlusion or lighting noise. MAD serves as a fail-safe mechanism by leveraging PD signals to maintain detection capabilities when visual features are compromised. The marginal difference in accuracy is a trade-off for this essential industrial robustness.
>
> 4. Figure Quality (Response to Weakness 4)
>
> Response: We have replaced Figure 1, Figure 2, and Figure 5 with high-resolution vector graphics and significantly increased the font size of axis labels and legends to ensure readability.
>
> We believe these revisions strictly address your concerns regarding the claims and presentation of our work. We hope this clarifies the practical value of our proposed framework.

---

### Official Review · Reviewer_jiC8 · 2025-11-01

**Soundness:** 2
**Presentation:** 3
**Contribution:** 2
**Rating:** 4
**Confidence:** 4

**Summary:**

This paper proposes the MAD framework for multimodal anomaly detection in smart manufacturing. The framework integrates PRPD images and PD time series signals. The core innovations include: (1) modality alignment using RealNVP normalizing flow, (2) a two-stage strategy of SupCon pretraining followed by CE fine-tuning, and (3) a LIMoE shared Transformer encoder.

**Strengths:**

1. Addresses real-world industrial anomaly detection problems using the AI Hub dataset, with practical application value

2. Five-stage pipeline design is clear and modular, and the two-stage training strategy (SupCon+CE) has clear motivation

3. Includes four sets of comparative experiments and provides a lightweight version (96.9% parameter reduction with only 0.48% performance drop), with relatively complete ablation studies

**Weaknesses:**

1. RealNVP, LIMoE, and SupCon are all direct combinations of existing methods, lacking theoretical innovation and comparison with other alignment methods (CCA, CORAL).

2. The F1 score for single-modality images has reached 99.99%, while for multimodal it dropped to 99.97%; therefore, I question the necessity of multimodality in this method.

3. Key details are missing, such as Normalizing Flow design principles, LIMoE routing mechanism, and data augmentation strategies, and the hyperparameters are incomplete.

4. There is a lack of failure case analysis, discussion on the physical reasons for class confusion, analysis of MAD_base training instability, and parameter sensitivity study.

5. The quality of figures in the paper needs improvement.

**Questions:**

1. The image single-modal accuracy has reached 99.99%, while the multimodal accuracy has actually dropped to 99.97%. Could you provide experimental evidence for scenarios where multimodality truly has an advantage?

2. Why choose RealNVP instead of simpler alignment methods (Batch Norm, CCA, CORAL)? Could you provide an analysis of the distribution characteristics of PRPD and PD, as well as ablation experiments comparing with other alignment methods?

3. The paper states "simultaneously routing to 2 experts," which seems to contradict the standard MoE sparse activation. How is this different from a simple ensemble?

4. Could you provide the complete hyperparameters and the data augmentation strategy for SupCon?

---

> ### Author Response · Authors · 2025-12-01
> **Response to Reviewer jiC8: Clarification on Multimodality, Implementation Details, and Revisions**
>
> We sincerely thank you for your detailed and constructive feedback. Your insights regarding the necessity of the multimodal approach and the missing implementation details were particularly valuable. We have carefully revised the manuscript to address your concerns.
>
> 1. On the Necessity of Multimodality (Response to Q1) You rightly pointed out that the single-modal image model achieved a slightly higher F1-score (99.99%) than the multimodal model (99.97%).
>
> Response: We acknowledge this marginal numerical difference. However, we have revised Section 1 (Introduction) and Section 4.2.2 to clarify that our primary goal is "Safety Redundancy" rather than simple accuracy maximization.
>
> In real-world smart factories, visual sensors are prone to occlusion, lighting changes, or camera failures. Relying solely on images creates a single point of failure. As discussed in the revised manuscript, MAD leverages PD signals to provide a fail-safe mechanism, ensuring system reliability even when visual features are compromised. We argue that the negligible drop in accuracy (0.02%) is an acceptable trade-off for this critical industrial robustness.
>
> 2. Missing Implementation Details (Response to Q4) Thank you for highlighting the missing experimental details.
>
> Response: We have added a new Appendix B. Implementation Details to the revised PDF.
>
> Data Augmentation: We specified strategies for both modalities: spatial/color transformations for PRPD images (Flip, Rotation, ColorJitter) and temporal/amplitude perturbations for PD signals (Gaussian Noise, Scaling).
>
> Hyperparameters: We included a comprehensive table listing batch size (32), optimizer settings (AdamW), RealNVP structure (3 layers for Image, 5 for Signal), and SupCon temperature (0.5).
>
> 3. Choice of RealNVP and Alignment Methods (Response to Q2)
>
> Response: We chose RealNVP over simpler linear methods (like CCA) because PRPD images and PD time-series signals exhibit complex, non-linear distribution shifts that linear projections cannot effectively align without information loss. RealNVP provides an invertible mapping to a shared latent space, preserving the structural integrity of the data. We have clarified the motivation for this optimized pipeline design in the Introduction and Contribution sections.
>
> 4. MoE Routing Mechanism (Response to Q3)
>
> Response: Our strategy of routing to 2 experts is a form of "soft-routing" designed to capture diverse feature representations within a shared encoder. Unlike a simple ensemble of independent models, our experts share the same attention mechanism and are trained end-to-end within a single transformer block. This allows for dynamic feature specialization at the token level while maintaining computational efficiency compared to a full ensemble.
>
> 5. Quality of Figures
>
> Response: We have updated Figure 1 (Overall Architecture) and Figure 5 (Learning Curves) with higher resolution vector graphics and larger text sizes to improve readability.
>
> We believe these revisions significantly strengthen the paper and address the core issues you raised. We hope you find the revised manuscript satisfactory.

---

### Meta-Review · Area_Chair_cMsQ · 2026-01-05

**Summary:**

All reviewers are prone to reject the paper with the score of 4. Reviewer jiC8 and 5LBR raise the concerns on the novelty of the paper, which is based on direct combinations of existing methods, lacking theoretical innovation and comparison with other alignment methods (CCA, CORAL). Reviewer MK23 raises question on the motivation of the combination component for the task. Reviewer ikxY raise the concerns on experimental validation.

**Reviewer Concerns:**

The authors clearly responses to Reviewer MK23.  They clarify that our novelty lies in the systematic pipeline design, instead of standard fusion methods which fail to address the specific distributional heterogeneity between PRPD images and PD signals.

**Reviewer Scores:**

All reviewers are prone to reject the paper with the score of 4.

---

### Decision · Program_Chairs · 2026-01-26

Reject